# The Coracohumeral Ligament and Its Fascicles: An Anatomic Study

**DOI:** 10.3390/jfmk10020149

**Published:** 2025-04-27

**Authors:** Emilio González-Arnay, Isabel Pérez-Santos, Camino Braojos-Rodríguez, Artimes García-Parra, Elena Bañón-Boulet, Noé Liria-Martín, Lidia Real-Yanes, Mario Fajardo-Pérez

**Affiliations:** 1Department of Basic Medical Sciences, Division of Human Anatomy and Embryology, University of La Laguna, Campus de Ofra s/n, 38320 Canary Islands, Spain; artimes23@gmail.com (A.G.-P.); alu0101138619@ull.edu.es (L.R.-Y.); 2Department of Anatomy, Histology and Neuroscience, Autonomous University of Madrid, 28049 Madrid, Spain; isabel.perezs@uam.es (I.P.-S.); camino.braojos@uam.es (C.B.-R.); 3MundoFisio-El Hierro, 38911 Canary Islands, Spain; 4Hospital Ship ‘Juan de la Cosa’, Social Institute of the Marine, 39009 Santander, Spain; elenabanonboulet@gmail.com; 5Division of Pathology, Canary Islands University Hospital, 38320 Canary Islands, Spain; nlm7589@gmail.com; 6Technical Department, Canarian Network of Pathology, 38206 Canary Islands, Spain; 7UltraDissection, 28049 Madrid, Spain; mfajardoperez@yahoo.com

**Keywords:** anatomy, coracohumeral, ligaments, rotator cuff, shoulder

## Abstract

**Background:** The coracohumeral ligament (CHL) is inserted in the coracoid process, from which it extends laterally and caudally, blending with the tendinous insertions of the subscapularis muscle and the supraspinatus muscle, with a third intermediate area between the muscles inserted between the humeral tubercles, and it contributes to the fibrous tunnel that engulfs the long head of the biceps tendon. Most previous studies mention insertions from the base of the coracoid process, but not from the tip, and some authors describe anterior and posterior columns. In contrast, others stress the existence of superficial and deep fascicles. Also, the relationship between the coracohumeral and the glenohumeral ligaments is unclear. Given the position of the CHL covering most of the rotator interval, and its role in the stability of the shoulder capsule and pathologies like frozen shoulder, a clear description of its fascicles in a plane-wise manner might be helpful for a selective surgical approach. **Methods**: We studied sixteen soft-embalmed shoulders to avoid misclassifying fascicles due to formalin-linked tissue amalgamation. Further histological assessment was performed on the two remaining non-embalmed shoulders. **Results**: In our sample, the coracohumeral ligament hung from the anterior and posterior edges of the coracoid process’ inferior surface, defining two columns that converged near the tip of the coracoid process. Both columns were formed by superficial and deep fascicles directed to different depths of the rotator cuff, usually via the rotator interval, fusing with the connective tissue around the muscles without direct distal attachments. We performed histological and morphometrical assessments, and we discuss clinical and biomechanical implications. **Conclusions**: The coracohumeral ligament contains four fascicles that fuse with the connective tissue of the shoulder joint, forming a double necklace around the subscapularis and supraspinatus. Therefore, its functions probably extend beyond simple vertical stabilization.

## 1. Introduction

The coracohumeral ligament (CHL) is classically described as medially inserted in the outer margin of the coracoid process. It reaches the greater and lesser tubercles, blending with the tendinous insertions of the subscapularis muscle (SScM) and the supraspinatus muscle (SpM), with a third intermediate area between the muscles that are described as inserted between the greater and the lesser tubercle of the humerus, therefore contributing to the fibrous tunnel that engulfs the long head of the biceps tendon (LHBBT). It also forms an essential part of the rotator interval, covering and merging with the connective tissue coating the rotator cuff muscles. The coracohumeral ligament has been described as essential for the stability of the shoulder joint [1,2,3,4,5], particularly in its cephalocaudal dimension; it is also a dynamic and flexible structure that is placed under tension in flexion, allowing a greater angle of flexion in medial rotation, and avoiding friction between the humeral head and the coracoid process itself by tightening and producing an eccentric movement of the humeral head during external rotation [6]. Thus, the CHL is not a mere structural conjoining between shoulder bones, but is functionally intertwined with the rotator cuff muscles. This is clearly shown in some studies that find the CHL to transfer the forces of supra- and infraspinatus muscles to the humerus, partially avoiding shoulder instability after rotator cuff tears [6,7,8]. This function is also related to the crescent-like tissue that runs perpendicular to the axis of tendons of the rotator cuff, and has been alternatively named the rotator cable [7], semicircular ligament of the humerus [9], circular fiber system [10], or transverse band [11]. However, few studies focus on anatomical descriptions. Also, many anatomical studies describe different fascicles of the CHL arranged in distinct layers [1,2,11,12,13]. This layered structure is also found in the rotator interval and shoulder joint capsule reinforcements [14]. However, the relationship between these structures and their morphometry has been scarcely explored in the literature [1,2]. In this work, we describe, fascicle-wise, the morphology of the fascicles of the CHL, and provide morphometrical data on their insertions in soft-embalmed cadavers.

## 2. Materials and Methods

Eighteen shoulders from seventeen cadavers (see Table A1) were used in this study (eight males and nine females, with a mean age of 76.9 ± 7.9 years old). Human tissue was obtained from volunteer body donors to the Applied Anatomy Laboratory at the Autonomous University of Madrid and the Department of Basic Medical Sciences of the University of La Laguna. Upon admission, cadavers were either subjected to immediate perfusion with Thiel solution or stored at −20 °C (case 17). Thiel solution [15,16] was prepared as a mixture of 14.3 L of a water-based solution containing 3 g BO_3_, 30 mL ethylene glycol, 20 g NH_4_NO_3_, and 5 g KNO_3_, and 500 mL of another solution consisting of 20 mL ethylene glycol and 1 mL 4-Cl 3-methyl phenol in 497 mL of water. Perfusion was transfemoral. The selection of upper limbs for dissection and further examination was random, and from cadavers that had undergone other experimental surgical procedures anatomically unrelated to the upper limb. Then, the upper limb was separated from the trunk with a MaDo^®^ Selekta 3(Maybachstraße 1, 72175 Dornhan, Germany) band saw, detaching it from a position slightly medial to the deltopectoral groove.

For dissection, the deltoid muscle was detached from its insertions (starting from the humeral ones), as well as the pectoralis major, the coracobrachialis, and the medial head of the biceps brachii. This process resulted in the exposure of the SScM and SpM muscle and any connective fascicle directed from the coracoid process to the humeral area superficial to the muscles above. In case 17, black ink (QPath^®^ 01816200 from Avantor 100 Matsonford Rd #200, Radnor, PA 19087, USA) was added to these connective fascicles. After exposing the superficial fascicles, a photographic register of each case was performed, and measures were taken using an LCD^®^ Vernier digital caliper (13979 SW Millikan Way, Beaverton, OR 97005, USA). For further dissection, a vertical cut was performed along the subscapularis and supraspinous fossae to detach both the SScM and SpM for their medial insertions, before gently pulling them laterally and dorsally. Once exposed, any connective tissue linking the area surrounding the humeral proximal epiphysis deeply to the muscles and the coracoid process was carefully dissected and followed towards its insertions. Again, the process was photographed and measured using a digital caliper. In case 17, blue ink was employed to stain these deep connective fascicles (QPath^®^ 01816203). Both shoulders of case 17 (not measured due to a divergent conservation method) were subjected to further processing, sampled in vertical slices encompassing the full cephalocaudal extent of the rotator cuff. Tissue blocks were tightly attached to a cork sheet (to minimize retraction artifacts) and immersion-fixed in 4% formaldehyde for 24 h before further sampling, inclusion in a histological cassette, dehydration, and paraffin treatment. The paraffin-embedded blocks were cut to a thickness of 5 µm with a rotating microtome, and then collected in a flotation bath with albumin-pretreated slides. After this, slides were kept on a hot plate at 40 °C. The samples were deparaffinized by exposing them to 60 °C in an oven for one hour, followed by overnight incubation in xylol and rehydration. Staining was performed using an automated hematoxylin–eosin kit, Sakura Prisma^®^ (76318-762, Flemingweg 10a, 2408 AVAlphen aan den Rijn, The Netherlands). Numerical analysis and descriptive statistics were performed using IBM SPSS^®^ 19 software. All procedures described here were carried out according to Spanish law and the Helsinki Declaration, and in the context of a body donation program approved by the respective boards of the University of La Laguna and the Autonomous University of Madrid, as well as being part of a broader project that received approval by the local ethics committee (reg. CHUC_2023_11.). The authors hereby confirm that every effort was made to comply with all local and international ethical guidelines and laws concerning the use of human cadaveric donors in anatomical research.

## 3. Results

The coracohumeral ligament is divided into an anterior (anteromedial, CHLam, Figure 1, Figure 2, Figure 3 and Figure 4) column and a posterior column (posterolateral, CHLpl, Figure 5 and Figure 6), with two fascicles each. These two columns converge in their proximal insertion near the tip of the coracoid process, forming a V-shaped structure with the vertex pointing laterally. Both fascicles of the CHLam are inserted in the inferior aspect of the coracoid process at different depths. They are directed towards the superficial and deep aspects of the SScM tendon. The posterolateral column of the CHL is inserted medially in the posterior edge of the caudal surface of the coracoid process, and laterally fuses with the capsuloligamentous tissue of the shoulder joint capsule: its most lateral fibers are directed towards the rotator interval, where they superficially cover the lateral area of the SpM tendon, while its most posterior and medial fibers are directed deeply towards the tendon of the SpM itself, before fusing with the tissue of the capsuloligamentous complex at this level.

The anteromedial column (CHLam) is differentiated into two fascicles; it is necessary to gently dissect the more superficial fascicle to access the deep one, which is located deep in the subcoracoid space. The superficial fascicle of the CHLam (Figure 1A,B,D,E, Figure 2A–C,E, Figure 3A,C,D,F,G and Figure 4A,B,D) is invariably attached to the caudal/inferior surface of the free portion of the coracoid process (1.23 ± 0.37 cm, ranging from 0.3 to 1.8 cm), with its insertion occasionally associated with the proximal insertion of a rather ventral coracoglenoid ligament (case 8, Figure 3A,B; see also case5 in Figure 2A), or reinforced (case 12, Figure 4A,B). Only exceptionally do the attachments of the superficial fascicle reach the coracoid base. In 43% of the cases (7/16), the CHLam was directed towards the superficial aspect of the myotendinous junction of the SScM, and merged with the overlying connective tissue around its insertion in the lesser humeral tubercle (e.g., Figure 1B,D,F). In the rest of the cases (9/16, 56%), this superficial fascicle ran directly to the rotator interval, fusing with connective tissue superficial to the subscapularis muscle after converging with the cephalic edge of the muscle (e.g., Figure 2B,F and Figure 3B). In one of the cases (5, see Figure 2A,B), it was possible to identify apparent insertions in the greater humeral tubercle. The longest dimension of the convergence between the CHL and the connective tissue was 1.25 ± 0.42 cm, ranging between 0.5 and 2.2 cm.

All the studied cases presented a deep fascicle of the CHLam (Figure 1B,C,E,F, Figure 2B,D,F, Figure 3B,C,F,G, and Figure 4C,D). In 7/16 cases (43.7%), it was found attached to the lateral aspect of the coracoid base. Two cases (2/16, 12.5%) presented additional insertions on the caudal surface of the coracoid process (e.g., 5, see Figure 2A,B and Figure 4D). Another four of the deep fascicles (1, 2, 14, 16, i.e., 25% of cases, see Figure 1A–D) were inserted slightly more rostrally, next to the anterior surface of the coracoid base itself, i.e., on the lateral anterior surface of the coracoid base. Three (7, 11, 12; Figure 2E,F and Figure 3F,G) of the deep fascicles were inserted into the angle between the inferior surface of the coracoid process and the base. Overall, this deep fascicle of the CHLam had some tendency to show a complex appearance, with multiple proximal insertions in different regions of both the base of the coracoid process and its free surface (6, Figure 2C,D), or even insertions in the deep face of the posterior column of the coracohumeral ligament in two cases (309, 13, see Figure 3F,G and Figure 4C). Two of the cases (3, 5 see Figure 1E and Figure 5A,B) showed an association between the insertion of the deep fascicle of the CHLam and the insertion of a coracoglenoid ligament, while in another three cases (2, 3, 11 Figure 1C,D and Figure 3G), the deep fascicle of the CHLam acted as a fascicle of the SGHL. The deep fascicles were mainly directed to adjoin the capsuloligamentous complex of the shoulder (10/16, 62.5% of cases), sometimes reaching it after entering the rotator interval (5/16, 31% of cases). The proximal insertions of the deep fascicle measured, along its longest axis, 0.71 ± 0.29 cm, ranging from 0.5 to 1.6 cm. The linear measurements of their lateral convergence with the capsuloligamentous tissue were 0.67 ± 0.32, ranging between 0.3 and 1.5 cm (see Table 1).

The posterolateral column (CHLpl) showed less variability than the anteromedial one. It was usually composed (proximally) of a single band of tissue that was inserted proximally either in the caudal surface of the coracoid process, close to its posterior edge (87.5%, 14 of 16 cases), or in its posterior edge itself (12.5%) (1.78 ± 0.45 cm; range: 1.1 to 2.8 cm). Distally, the CHLpl fused with the capsuloligamentous tissue of the shoulder joint capsule at the level of the insertion of the supraspinatus in the greater tubercle of the humerus. A superficial and a deep fascicle could also be distinguished in the CHLpl: the most lateral fibers of the CHLpl merged superficially to the SpM tendon (Figure 5A,C–E,G and Figure 6A–C,H; insertion length: 1.38 ± 0.44 cm, in a range from 0.3 to 2.2 cm) and formed the superficial fascicle, while the most posterior fibers were directed towards the greater tubercle and merged with the connective tissue beneath the SpM tendon (insertion length: 0.46 ± 0.28 cm, in a range from 0.2 to 0.4 cm, in the cases in which it was possible to isolate it; see below), forming the deep fascicle.

In most cases (12 of 16, 75%), the deep fascicle was composed exclusively of the fibers arising in the posteromedial insertions of the CHLpl itself. In contrast, in one case (6, Figure 6D), there was a distinct auxiliary fascicle arising from the lateral border of the suprascapular notch and directed towards the capsuloligamentous complex below the SpM tendon, being the only deep structure of the posterolateral column. In two other cases (5, 6; see Figure 6A,B,I), this auxiliary fascicle was closely related to, or even a reinforcement of, the main fascicle of the CHLpl. There were also some minor variations in the lateral aspect of the CHLpl, which was bifurcated in one case (1, Figure 5B), and reached its target directly through the rotator interval in another single case (5, Figure 6B). One of the cases (case 15) showed a very close relationship between the most superficial fibers of the posterolateral column and the coracoacromial ligament. In case 8, the deep fascicle was not identified. Regarding linear measurements of the insertions of the CHLpl, it was difficult to isolate the superficial fascicle from the deep fascicle, as the transition was gradual. It followed the shape of the posterior edge of the coracoid, smoothly embracing the upper border of the SpM. As far as it was possible to isolate the proximal insertions of the deep fascicle, they measured 0.52 ± 0.27 cm (range: 0.2 to 0.9 cm). When the transition was too smooth, the deep fascicle was recorded with the superficial one, and registered in Table 1 as Incl. (standing for ‘included’).

Careful dissection of the lateral insertions of the CHL showed a rather particular multi-layered (Figure 7 and Figure 8) structure that did not directly attach to any of the humeral tubercles as classically described, except in one case (see above and Figure 2A,B). Instead, fascicles were incorporated into a ligamentous cuff (recorded as capsuloligamentous complex) composed, at least, of the fascicles of the CHL themselves and of the connective (fascial) tissue surrounding the muscles of the rotator cuff. This convergence occurred in successive planes that were, in turn, related to the relative positions of the proximal insertions of the CHL (Figure 9). The most superficial layer, when present, corresponded to the most lateral end of the CHL (pl) and its insertion in the rotator interval (see arrow in Figure 8A), which fused itself with the most superficial tissue of the capsuloligamentous wall of the shoulder joint capsule and covered the LHBBT (discontinuous arrow in Figure 8A). This superficial layer was closely attached to a second layer originating from the superficial fascicles of the CHL (am) and the CHL (pl). It formed a continuous fascicle that fused into the connective tissue overlying the SScM and the SpM. The third, deeper layer was generated by the fusion of the deep fascicle of the CHL (am) and the deep fascicle of the CHL (pl) that took place deep in the fascial tissue of the deeper aspect of both the SScM and the SpM (see Figure 7, Figure 8, Figure 9 and Figure 10).

## 4. Discussion

### 4.1. Technical Considerations

Here, we present a descriptive and morphometric study of the CHL in soft-embalmed shoulders using Thiel solution. We describe the CHL as a tent-like structure that hangs from the anterior and posterior edges of the coracoid process’ inferior surface, defining two columns (CHLam and CHLpl) that converge near the tip of the coracoid process, forming a V-shaped structure. Both columns are formed by superficial and deep fascicles directed to different depths of the rotator cuff, usually via the rotator interval. The deep fascicle of the CHLam usually originates deep in the subcoracoid space. Lateral/distal insertions of the CHL are not pure insertions, but rather variable areas of convergence between the CHL itself and the rest of the connective tissue of the shoulder’s capsuloligamentous tissue. The results here partially overlap with findings in the existing literature regarding anatomy and morphometry [1,2,3,4], albeit showing a much more complicated disposition. This difference in results is related to the difference in fixation methods, as the rigidity of formalin-embedded tissue increases structures’ fragility and amalgamates them, making their separation unfeasible [9,17]. The soft-embalmed tissue employed in this study offers a valid alternative, as it maintains the flexibility of in vivo tissue through perfusion of a hypertonic solution that, in turn, leads to the shrinkage of flexible structures along actual demarcation planes, facilitating the distinction between thin layers of tissue. The latter feature offers an advantage for studying layered connective structures like the CHL, in comparison to fresh or fresh-frozen body parts. The main shortcomings of this study are related to the design of cadaveric studies itself, as the structure of body donation programs does not allow researchers to retrieve subtle clinical details, like the presence of chronic shoulder pain, which may be prevalent in an aged population like the one represented in this kind of study. Still, our findings refer to an actual population.

### 4.2. Anatomical Considerations Regarding the Layered Structure of the CHL

A relevant interpretation of the results of the present study is that the CHL is not a ligament in its classical sense [12], as it does not attach directly as a single fascicle to bony structures, exerting its function as a reinforcement of a broader connective system with its own set of insertions [7,13]. In this sense, the CHL plays a role in the structure of the shoulder joint, providing both its most superficial layer (superficial fascicles of CHLam and CHLpl) and an intermediate layer immediately overlying the glenohumeral ligaments formed by the deep fascicles of the CHLam and CHLpl (Figure 11). This second layer, although recognized by many other authors [1,2,13], has been described as belonging to the same plane as the glenohumeral ligaments, a finding that is somewhat inconsistent with the function of the CHL as a clamp for the rotator interval muscles (see below), and is probably related to the different fixation methods used (e.g., 8% formalin [2], see above), as well as to the limitations inherent to shoulder imaging techniques [18]. Although we found some cases of apparent convergence between the CHL and the coracoglenoid or SGHL, they did not represent the canonical organization of this structure in our sample. The disposition of the proximal insertion of the CHL also points to its roles as anterior, posterior, and even caudal walls of the subcoracoid space. The involvement of the deep fascicles of the CHL in subcoracoidal impingement syndromes may explain the synchronous apparent swelling of the MGHL [18].

Regarding the deep fascicle of the CHLam, further studies are needed to understand the role of its somehow oblique trajectory, but it may be related to a role in the stability of the bursae. Interestingly, the disposition of the deep fascicle of the CHLam, in some cases, matches the one depicted for the putative suspensory ligament of the bursa [19], and both structures might be the same. Also, the deep fascicle of the CHLam may be at least the most superficial aspect of the arthroscopic comma sign tissue [20].

### 4.3. The Role of the CHL as an Essential Contributor to the Rotator Cable and Its Clinical Correlate

Thus, the findings presented here reinforce the idea of the CHL as a multi-layered structure that merges into the multi-layered nature of the shoulder capsule [14], contributing to both its most superficial layer and one of its intermediate-deep layers (see below and Figure 9, Figure 10 and Figure 11, and particularly 11, and probably indirectly forming two different necklaces or crescents (from the superficial CHLam to the superficial CHLpl, encompassing the SScM tendon, and from the deep CHLam to the deep CHLpl, encompassing at least the SpM tendon) around the rotator cuff; this is related to the concept of the semicircular ligament of the humerus [9], or the rotator cable [7]. Therefore, the CHL is not a mere chord of tissue from which the rotator cuff muscles or the humeral head hang, but rather a complex system that holds together the rotator cuff muscles by embracing both their superficial and deep aspects and limiting their movement regarding the coracoid process (therefore acting as a brace or clamp). This ‘clamping’ function is illustrated by recent results [8] that demonstrate the ability of the shoulder joint to work without instability, even in the presence of extensive rotator cuff tears, which would be explained by the continuing function of the CHL holding the remaining muscles (as long as it remains intact). As initially described, the distal insertions of the CHL and the bridging tissue between them are also a central part of the rotator cable [7]. Overall, the data suggest a role of the CHL not only as a ‘control’ of external rotation and inferior displacement of the joint, but also of internal rotation and anterior displacement [21,22,23]. This control is exerted in the whole mediolateral and anteroposterior dimensions of the insertions of the rotator cuff muscles, as there is a high degree of symmetry between the CHLam and CHLpl fascicles when they are identifiable as individualized structures, and they show a similarly symmetric behavior when they distally fuse into a broader connective tissue complex. Indeed, regardless of whether the CHL is still identifiable, the shoulder joint always has at least four layers [14,24]. Also, these biomechanical properties provide the CHL with a role in the treatment of any kind of shoulder instability, not just inferior and multidimensional instability, which is a relevant result, as the CHL is an extra-articular structure that is better suited to open approaches [25]. The lack of a fascicle-wise approach in present surgical techniques may explain, to some degree, their inconsistency regarding adverse outcomes (see Karovalia et al., 2019, for a review) [24]; in this sense, the development of fascicle-wise approaches may be helpful for less invasive treatment of shoulder instability, resulting in fewer cases of movement limitations than rotator interval closure. This approach may also be helpful for focused surgical treatment of subcoracoid impingement syndrome [26].

## 5. Conclusions

The coracohumeral ligament is composed of two columns of connective tissue originating in the coracoid process’s inferior surface. Each column contains a superficial and a deep fascicle. The deep anterior fascicle sometimes coincides with what has been termed the suspensory ligament.

Superficial fascicles of the CHL fuse with the connective tissue overlying the rotator interval muscles, while deep fascicles fuse with the connective tissue underlying them. Altogether, they form a double necklace, mostly around the SScM and SpM.

There are no direct lateral insertions of the CHL fascicles into bony structures, but there is a progressive fusion into a wider and multi-layered cuff of connective tissue formed by the glenohumeral ligaments, the connective tissue surrounding the rotator cuff muscles and the coracohumeral fascicles.

## Figures and Tables

**Figure 1 jfmk-10-00149-f001:**
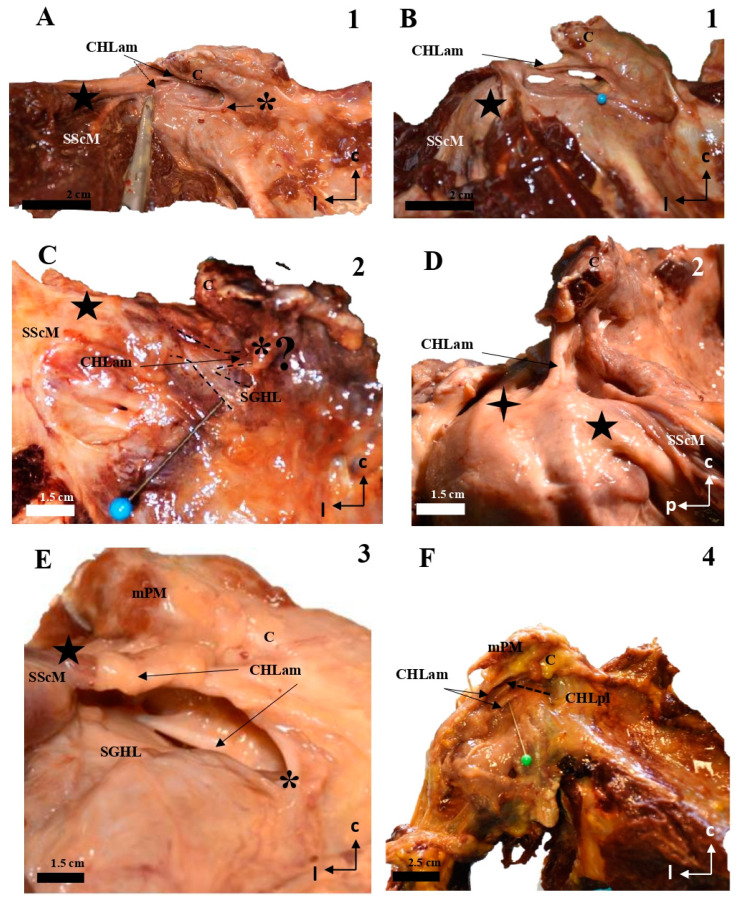
Cadaveric dissections of the anterior column of the CHL. (**A**,**B**) (case 1): after disinsertion of the SScM, it was laterally retracted, revealing the anterior and medial column of the CHL (CHLam) divided into two fascicles: a superficial one that attached to the caudal surface of the coracoid process (C) (arrow in **A**,**B**) and the capsule-ligamentous tissue of the humeral head, positioned above the tendon of the SScM; and a smaller deep fascicle attached to the anterolateral aspect of the base of the coracoid process (dotted arrow in **A** and pin in **B**) and the capsuloligamentous tissue that covered the humeral head, located beneath (deep into) the SScM tendon (five-pointed star). In (**B**), the shoulder is in forced external rotation. (**C**,**D**) (case 2): In this shoulder, the CHLam exhibits a single superficial fascicle fully attached to the caudal surface of the coracoid process (**D**), while the deep fascicle is replaced by a loose band of tissue that fuses with the SGHL (**C**) and may correspond to an atypical coracoglenoid ligament based on its insertions (asterisk and dotted lines). The superficial fascicle (**D**) laterally fuses with the shoulder capsule at the rotator interval (located between the SScM tendon, marked with a five-pointed star, and the SSpM tendon, marked with a four-pointed star). In cases 3 (**E**) and 4 (**F**), the superficial fascicle attaches to the caudal surface of the coracoid and fuses with the shoulder capsule above the SScM tendon, while the caudal fascicle merges with the capsule beneath the SScM tendon, originating from the lateral aspect of the coracoid base (dotted arrow in **F**). The asterisk in (**E**) highlights a distinct coracoglenoid ligament.

**Figure 2 jfmk-10-00149-f002:**
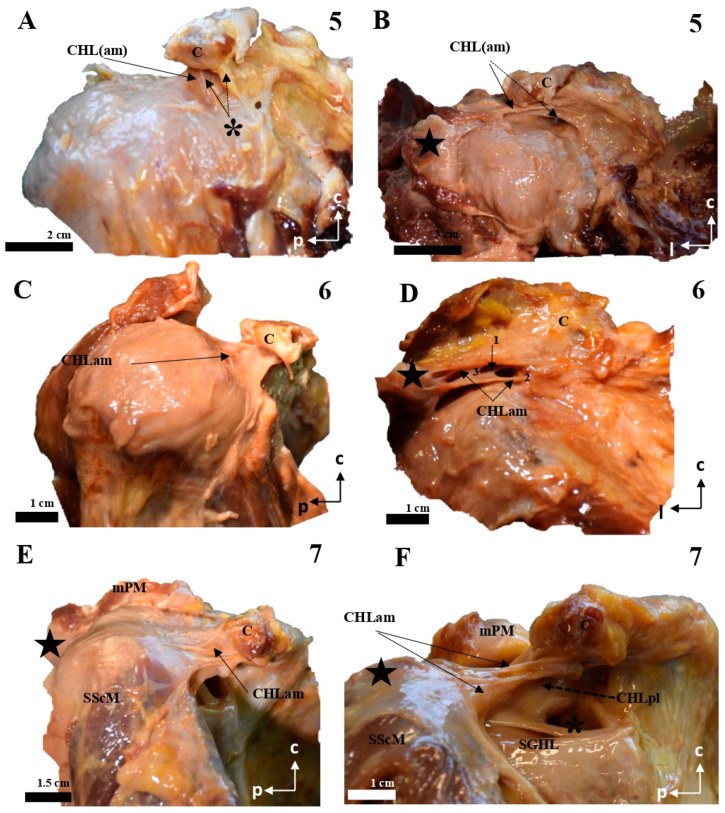
Cadaveric dissections of the anterior column of the CHL. (**A**,**B**) (case 5): the anterior and medial column of the CHL (CHLam) is organized into two fascicles: a superficial one that is inserted in the caudal surface of the coracoid process (C) (arrow in **A**) immediately laterally to a lateral fascicle of a two-bundled coracoglenoid ligament (asterisk, the medial bundle marked with a dotted arrow), and broadly reaches the rotator interval; and a deep fascicle (dotted arrow in **B**) that is inserted in the caudal surface of the coracoid and the lateral aspect of its base, and adjoins the humeral capsuloligamentous tissue beneath the insertion of the SScM tendon (star in **B**). (**C**,**D**): Case 6 shows a similar pattern, although the deep fascicle (**D**) is organized into three bundles in its medial insertion: one directed to the posterior edge (1) of the coracoid base, a second one directed to the anterior edge of the coracoid base (2) and a third one merging with the inferior surface of the superficial fascicle (3). Case 7 shows a superficial fascicle ranging from the caudal surface of the coracoid process (**E**) to the rotator interval and the capsuloligamentous tissue of the humeral head superficial to the insertion of SScM, partially reaching the humeral tubercle. The deep fascicle is a strong band of connective tissue that arises from the most posterior and medial aspect of the caudal coracoid surface (**F**), whose lateral insertion is superimposed on the lateral fascicle of an accessory band of the SGHL (four-pointed star).

**Figure 3 jfmk-10-00149-f003:**
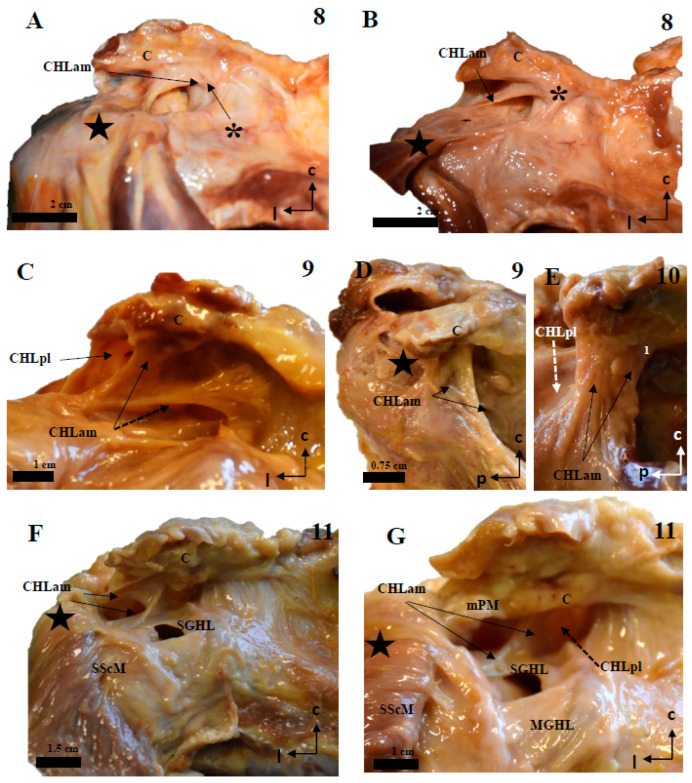
Cadaveric dissections of the anterior column of the CHL. (**A**,**B**) (case 8): the anterior and medial column of the CHL (CHLam) is organized into two fascicles: a superficial one that is inserted in the caudal surface of the coracoid process (C) (arrow in **A**), immediately laterally to a lateral fascicle of the coracoglenoid ligament (asterisk)—this superficial fascicle profusely fuses with the fascial tissue overlying the SScM tendon (star), before reaching the capsuloligamentous tissue of the shoulder; and a deep fascicle that is inserted in the caudal surface of the coracoid and the lateral aspect of its base, only revealed after careful removal of the superficial fascicle, and adjoins the humeral capsuloligamentous tissue beneath the insertion of the SScM tendon (star in **B**). The coracoglenoid ligament is present (asterisk in **B**), and is laterally prolonged towards the capsuloligamentous complex. Case 9 (**C**,**D**) shows two fascicles: the superficial one is broadly comparable to the one previously described (**C**); the deep fascicle (**C**,**D**) is a Y-shaped bundle that ranges from the lateral aspect of the coracoid base to the rotator interval (**D**) and the deep fascia of the SScM. Case 10 (**E**) is a good example of the canonical organization of the CHLam, with a superficial bundle that extends from the caudal surface of the coracoid to the rotator interval (cephalic and superficial to the SScM tendinous insertion) and broadly reaches the rotator interval, and a deep fascicle broadly extending from the lateral surface of the coracoid base (1), before fusing with the shoulder joint capsule underneath the SScM tendon. In case 11 (**F**,**G**) the deep fascicle of the CHLam is inserted almost in the posterior edge of the caudal surface of the coracoid limb, as well as in the anterior surface of the CHLpl (see dotted arrow in **G**), before reaching the shoulder joint capsule, as an accessory fascicle of the superior glenohumeral ligament (SGHL in **G**).

**Figure 4 jfmk-10-00149-f004:**
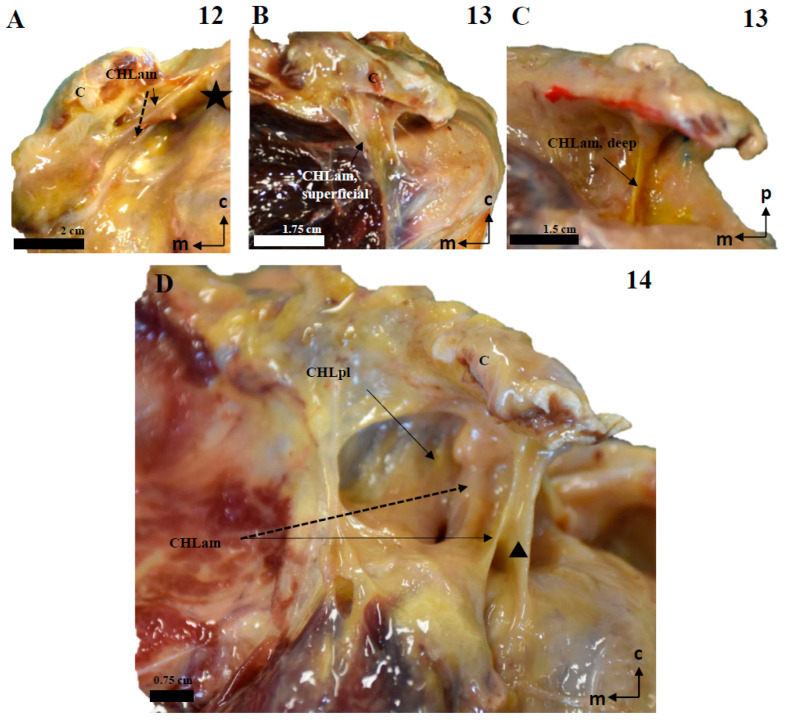
Cadaveric dissections of the anterior column of the CHL in shoulders, as seen from the left side. (**A**) (case 12): The anterior and medial column of the CHL (CHLam) is inserted in the caudal surface of the coracoid process and reinforced both laterally (dotted arrow) and medially (arrow), being the medial reinforcement at the anterolateral aspect of the coracoid base, where it fuses with the coracoid insertion of a deep fascicle (black star). Laterally, the major superficial fascicle and the deep fascicle surround the insertion of the SScM tendon. (**B**): Case 13 shows a superficial fascicle inserted medially at the caudal surface of the coracoid process (and partially in the lateral aspect of the base); this fascicle is laterally inserted in the superficial fascia of the SScM. The deep fascicle (yellow in **C**) is inserted in the inferior space of the coracoid limb and the anterior surface of the CHLpl. Note the differences in the insertions of both the superficial (red) and deep (yellow) bundles. (**D**) shows a particular distribution of the anterior column of the CHL; the superficial fascicle blends with the superficial fascia of the SScM near its lateral insertion. In this lateral view of the coracoid process, it appears (arrow) as an intermediate bundle situated next to a particularly strong reinforcement of the edge (triangle), between the CHLam and the CHLpl and the deep fascicle of the anterior column, which is inserted in the anterolateral aspect of the coracoid base and blends laterally with the deep fascia of the SScM insertion (dotted arrow).

**Figure 5 jfmk-10-00149-f005:**
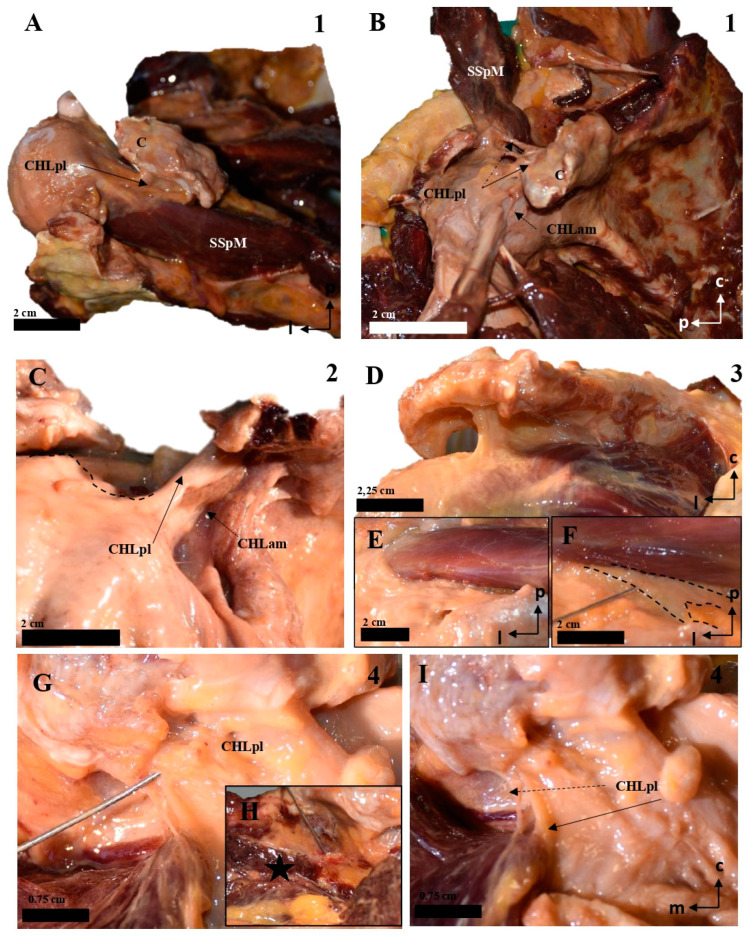
Cadaveric dissections of the posterior column of the CHL. (**A**,**B**) (case 1): The posterior and lateral column of the CHL (CHLpl) is organized into two fascicles: a superficial one that is inserted in the caudal surface of the coracoid process (C) (arrow in **A**) and fuses with the fascial tissue overlying the SSpM tendon (star), before reaching the capsuloligamentous tissue of the shoulder; and a deep fascicle, bifurcated in this particular case, which is simply the posteromedial continuation of the superficial fascicle; its coracoid insertions allow it to blend into the capsuloligamentous tissue of the shoulder joint beneath the SSpM (dotted arrow in **B**). This organization is repeated in cases 2 (**C**), 3 (**D**–**F**, dotted lines representing the outline of the structure), and 4 (**G**–**I**). In all of them, the most lateral bundles of the CHLpl blend with the capsule underlying the SSpM (highlighted in (**C**) with a dotted line), and the most medial and posterior bundles of the CHL (**E**,**F**,**I**) reach the capsule under the same muscle. Additionally, case 4 shows an auxiliary deep fascicle (dotted arrow in I) that arises in the posterior aspect of the scapular notch (**H**). Note that in images (**G**,**I**), the SSpM (star) has been sectioned and displaced posterolaterally, while in images (**E**,**F**), its insertions have been respected.

**Figure 6 jfmk-10-00149-f006:**
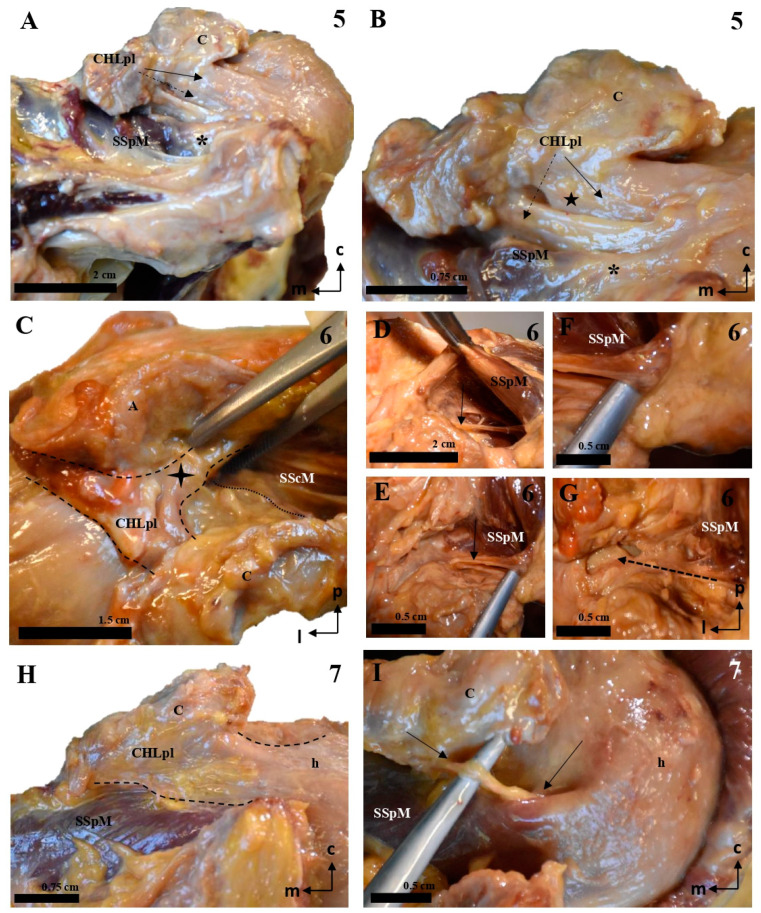
Cadaveric dissections of the posterior column of the CHL. (**A**,**B**) (case 5): The posterolateral column of the CHL (CHLpl) is inserted proximally in the posterior edge of the caudal surface of the coracoid process, and laterally, it fuses with the capsuloligamentous tissue of the shoulder joint capsule: its most lateral fibers are directed towards the rotator interval (solid arrow in **A**), where they superficially cover the lateral area of the SSpM tendon, while its most posterior fibers are directed deeply towards the tendon (dotted arrow in **A**). An auxiliary fascicle (dotted arrow in **B**) is inserted proximally in the posterior edge of the angle between the coracoid process and the coracoid base, and fuses with the capsuloligamentous tissue of the capsule (asterisk) underneath the SSpM tendon, albeit in a slightly more superficial plane than the posterior fibers of the main CHLpl (five-point star). (**C**–**G**) (case 6): almost all the fibers of the CHLpl are directed from the posterior edge of the inferior surface of the coracoid process (c) to the most superficial layer of the shoulder capsuloligamentous tissue (four-point star in **C**). Running parallel to the anterior/deep surface of the SSpM (**D**,**E**) appears an auxiliary fascicle that is proximally inserted in the posterior aspect of the scapular notch (**F**) and fuses with the deepest layer of the shoulder joint (**G**). Case 7 (**H**,**I**) follows a canonical pattern, with the most lateral fibers directed from the posterior edge of the inferior coracoid surface (c) to the rotator interval covering the insertion of the SSpM tendon (dotted line in **H**). A bundle of posteriorly inserted fibers envelops the SSpM tendon deeply (arrows in **I**).

**Figure 7 jfmk-10-00149-f007:**
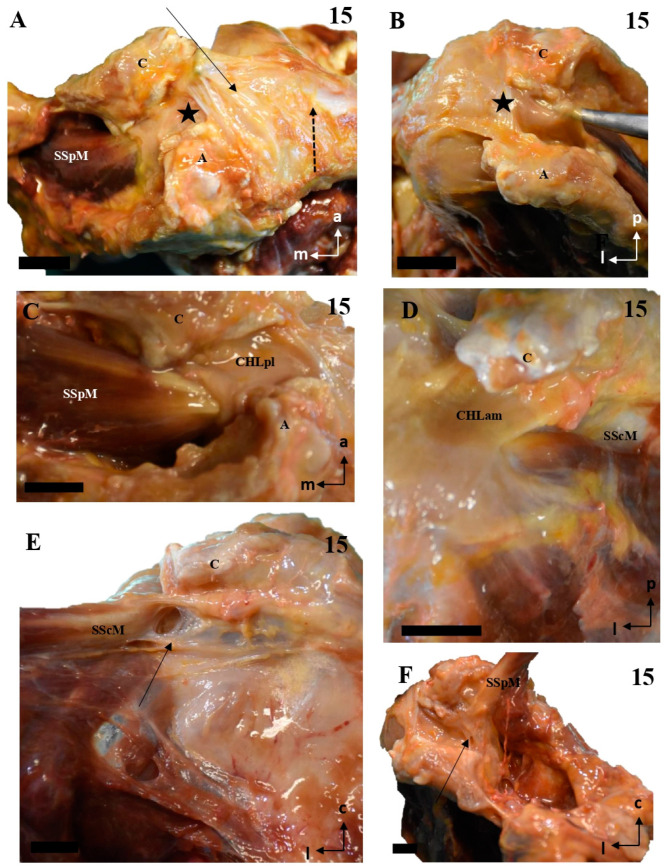
The multi-layered structure of the humeral insertion of the CHL. (**A**): The most superficial layer corresponds to the CHLpl and its insertion in the rotator interval (arrow), which fuses itself with the most superficial tissue of the capsuloligamentous wall of the shoulder joint capsule and covers the LHBBT (discontinuous arrow). Its most posterior and medial fascicle follows a coracoacromial route and is, indeed, attached to the inferior surface of the coracoacromial ligament (sectioned) (five-point star in **A**,**B**). (**B**–**D**): The superficial layer is partially intermingled with a second layer extending between the superficial aspects of the rotator interval muscles, connecting their tendons; this layer originates (or at least attached to) both from the anteromedial and posterolateral columns of the CHL, and effectively represents most of what is classically understood as the CHL. (**E**,**F**): The deepest macroscopic layer is a similar bundle that extends between the deep fascicles (arrows) of both columns, therefore reinforcing the deep attachment of the muscular tendons at the boundaries of the rotator interval. The scale bars represent 1 cm.

**Figure 8 jfmk-10-00149-f008:**
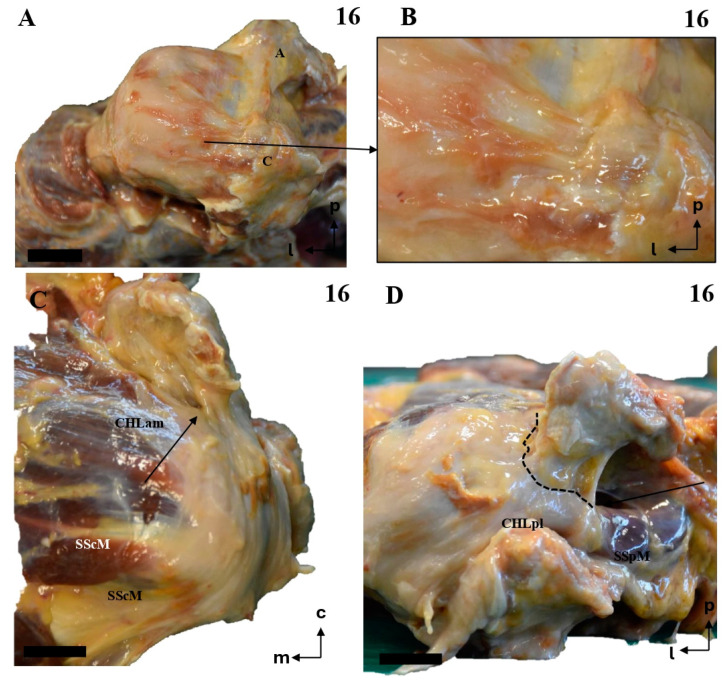
The multi-layered structure of the humeral insertion of the CHL. (**A**,**B**): The most superficial layer corresponds to the most lateral aspect of the CHL, where fibers around the apex of the V-shaped insertion along the caudal surface of the coracoid limb follow a superomedial-to–caudolateral route towards the rotator interval, before merging with the capsuloligamentous tissue of the shoulder in a superficial plane. (**C**,**D**): The main body of both the anterior and posterior columns of the CHL is inserted laterally in the rotator interval, partially overlapping (arrows) with the tendinous insertions of both the SScM (anterior column) and the SSpM (posterior column), effectively forming a single arch (dotted line in **D**) which links both muscles. The scale bars represent 1 cm.

**Figure 9 jfmk-10-00149-f009:**
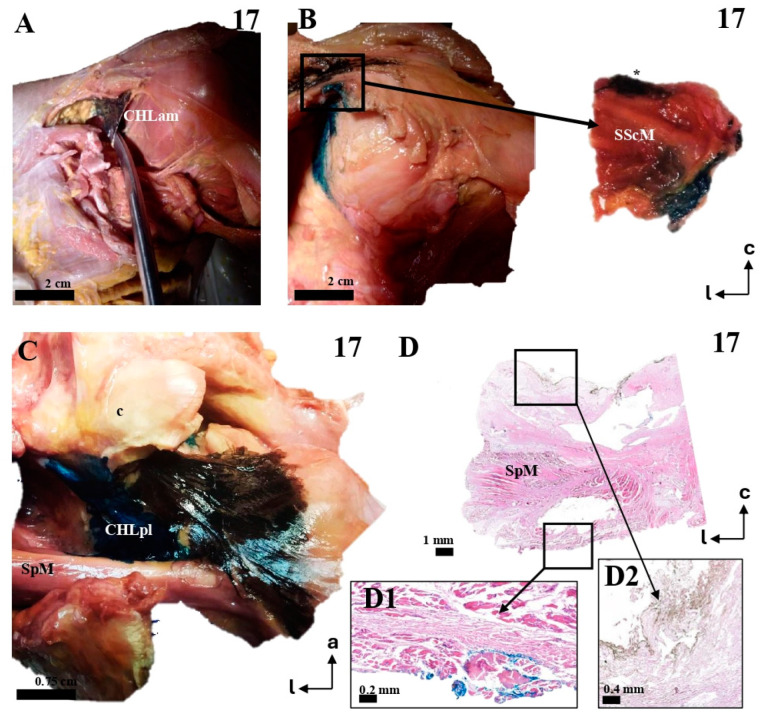
The multi-layered insertion humeral insertion of the CHL. (**A**,**B**): A frontal view of the subscapular region, with the superficial fascicle of the CHLam inked in black (**A**) and the deep fascicle inked in blue (**B**). The detail shows a macroscopic picture of a section of the SScM, demonstrating the distribution of the fascicles along its superficial (asterisk) and deep surfaces. (**C**): a zenithal view of the coracoid process (c) and the suprascapular fossa, with the SpM displaced posteriorly in order to view the disposition of the posterior fascicles (superficial in black, see **D2**, and deep in blue, see **D1**) around its myotendinous insertion. (**D**): A histological picture of the deep (**D1**) and superficial (**D2**) fascicles of the CHLpl around the SpM lateral portion.

**Figure 10 jfmk-10-00149-f010:**
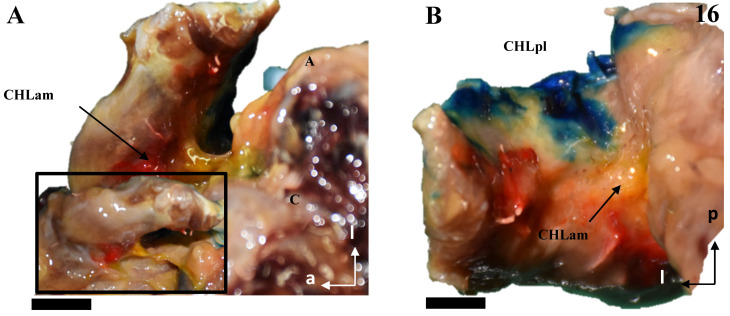
The multi-layered structure of the scapular insertion of the CHL. (**A**): The insertion of the superficial fascicle (red) of the anterior column (CHLam) overlaps mediolaterally with the insertion of the deep fascicle (yellow), which occupies the medial and inferior aspect of the coracoid base (see **B**). Both fascicles of the posterolateral column (CHLpl) are inserted in the posterior edge of the coracoid process (**B**). The rectangle in A depicts the process of liberating the deep fascicle of the CHLam during the dissection. The scale bars represent 0.5 cm.

**Figure 11 jfmk-10-00149-f011:**
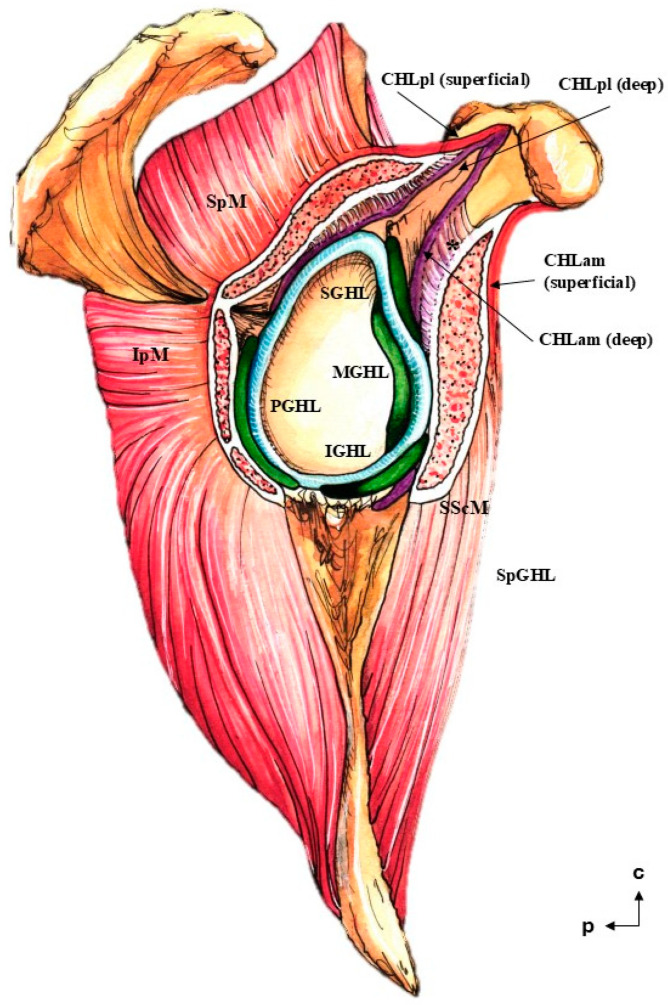
The putative distribution of the reinforcement of the shoulder joint capsule. The most superficial plane is composed of the superficial fascicles of both the CHLam and the CHLpl (red). The intermediate level (purple) is formed by the deep fascicles of both columns of the CHL, as well as the spiral glenohumeralligament (SpCHL). The deepest level is formed by the canonical glenohumeral ligaments: superior (SGHL), middle (MGHL), anteroinferior (IGHL), and posteroinferior (PGHL). Note the clamp formed by the fascicles of the CHL around the subscapularis (SScM) and supraspinatus muscles (SpM). The relationship between the CHL and the infraspinatus muscle (IpM) is unclear as far as the authors are concerned. Regarding the deep column of the CHLam, its set of proximal insertions is variable, and only one typology is depicted; also note its double function/overlap as a suspensory ligament of the subscapular bursa (asterisk).

**Table 1 jfmk-10-00149-t001:** Linear measurements of the CHL insertions in centimeters. Note that the distal measurements do not represent true insertions, but linear measurements of the convergence between the fascicle of the CH and the capsuloligamentous complex of the shoulder. The fascicles of the posterior column are difficult to discriminate, and in many cases, separate measurements are not reported (‘Incl.’ for ‘included’). n.e in case 1 stands for not evaluated, as it was either non-existent or artifacted during dissection ‘None’ was reserved for a case (8) where some of the expected structures were clearly absent.

Case	Anterior Column				Posterior Column			
	Superficial		Deep		Superficial		Deep	
	Proximal	Distal	Proximal	Distal	Proximal	Distal	Proximal	Distal
**1**	1.2	1.4	n.e	0.7	2.2	0.8	Incl	Incl
**2**	1	2.2	0.5	0.4	1.18	2.23	Incl	Incl
**3**	1.6	1.3	0.45	0.3	1.25	1.1	0.2	0.2
**4**	0.3	0.5	0.5	0.56	1.75	1.5	Incl	Incl
**5**	1	1	0.7	0.3	1.1	0.3	0.6	0.4
**6**	1.2	0.8	0.5	0.6	1.7	1.5	0.3	0.4
**7**	1.2	1.6	0.6	1.5	2.3	2	0.6	0.2
**8**	1.5	1.3	0.7	1	1.7	1.3	None	None
**9**	1	1.1	0.6	0.5	1.3	1.2	Incl	Incl
**10**	0.7	1.2	1	0.9	1.6	1.6	0.9	0.9
**11**	1.4	1	0.9	0.7	1.6	1.4	Incl	Incl
**12**	1.8	0.8	1.6	0.9	1.4	1.6	Incl	Incl
**13**	1.4	1.2	0.7	1	2.8	1.55	Incl	Incl
**14**	1.5	1.1	0.5	0.6	1.7	1.2	Incl	0.7
**15**	1.4	1.8	0.8	0.3	1.8	1.6	Incl	Incl
**16**	1.5	1.7	0.6	0.5	2.1	1.2	Incl	Incl
**Total**	1.23 ± 0.37	1.25 ± 0.42	0.71 ± 0.29	0.67 ± 0.32	1.78 ± 0.45	1.38 ± 0.44	0.52 ± 0.27	0.46 ± 0.28

## Data Availability

All data, including detailed photographical records of the dissections, are fully available on request to EGA or IPS. Most of the cadaveric material is still stored.

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
