# Peer review of "The Coracohumeral Ligament and Its Fascicles: An Anatomic Study"

_jfmk, 2025, doi:10.3390/jfmk10020149_

Round 1

Reviewer 1 Report

Comments and Suggestions for Authors

Thank you for the invitation to review the paper “The Coracohumeral Ligament and Its Fascicles: An Anatomic Study. In this study, González-Arnay et al. provide a comprehensive examination of the coracohumeral ligament, enhancing the understanding of its anatomical structure and role within the shoulder region. The authors observed the presence of two columns inserting near the tip of the coracoid process of the scapula. They also identified superficial and deep fascicles of collagen fibers in relation to the rotator cuff muscles, particularly the subscapularis and supraspinatus muscles. Furthermore, the study includes histological and morphometric assessments, offering additional insights into the coracohumeral ligament. Although the number of cases considered is not large, this study provides valuable anatomical and functional information on the coracohumeral ligament, with important implications for both surgical approaches and the management of shoulder joint pathologies. In my opinion, the authors should consider including Table S2 within the main text rather than as supplementary material.

Author Response

Comment 1:

Although the number of cases considered is not large, this study provides valuable anatomical and functional information on the coracohumeral ligament, with important implications for both surgical approaches and the management of shoulder joint pathologies

Response 1: 

Thank you for your kind review. We have made some changes to the text in order to improve the way in which we communicate our results. Most of them are repeated errors like 'auxiliary' instead of 'auxiliar'. There are several other minor corrections in lines 24, 55, 67-70, 93, 94, 128, 135, 136, 273-286 and 316-331, as well as the captions of figures 2,3,5,6,7. 

Comment 2

 In my opinion, the authors should consider including Table S2 within the main text rather than as supplementary material.

Response 2: 

Thank you for this suggestion. Done. 

Un saludo y gracias. 

Reviewer 2 Report

Comments and Suggestions for Authors

This is a most interesting paper. The anatomy of the CHL has always for many been difficult to understand.

The authors need to be complimented on their attention to detail in the anatomic preparation and dissections. The findings and results are well described and the English without fault.

Two minor comments; In  Fig.11, which is the key Fig. to the paper, the Arrow from the CHLam(deep) seems to go to the same structure as the arrow from the CHLpl(deep). Is this correct? In line 367 CHLa should perhaps be CHLam.

This is a very well done paper.

Author Response

Comment 1 In  Fig.11, which is the key Fig. to the paper, the Arrow from the CHLam(deep) seems to go to the same structure as the arrow from the CHLpl(deep). Is this correct?

Response 1: Indeed, it was an error that has been corrected. The compass was also modified. Thank you very much for this comment. 

Comment 2

Response 2: In line 367 CHLa should perhaps be CHLam. Already corrected, thank you. 

Thank you very much for your encouraging words. 

Un saludo. 

Reviewer 3 Report

Comments and Suggestions for Authors

This anatomical study examines the Coracohumeral Ligament (CHL) and its fascicles. The CHL originates from the coracoid process, extends laterally and caudally and blends with the subscapularis and supraspinatus tendons, forming a fibrous tunnel around the biceps tendon. Previous studies have varied in describing its structure, with views of anterior and posterior columns or superficial and deep fascicles. This study aimed to clarify the CHL’s fascicles through histological assessments on embalmed and non-embalmed shoulders. The results revealed four fascicles that fuse with surrounding tissue, forming a double necklace around the subscapularis and supraspinatus. This suggests that the CHL’s role in shoulder stability extends beyond vertical support. These findings have significant clinical and biomechanical implications for shoulder pathology and surgery. Overall, the article is well-written and provides valuable insights and I have no further suggestions for improvement.

Author Response

Thank you very much for your review and your kind words. We have included some modifications in the revised version, mainly dealing with scientific English, as well as some labeling errors in Figure 11. We hope that the revised manuscript improves the previous one.

Un saludo y gracias